# Moderate-to-Vigorous Physical Activity in Primary School Children: Inactive Lessons Are Dominated by Maths and English

**DOI:** 10.3390/ijerph18030990

**Published:** 2021-01-22

**Authors:** Andy Daly-Smith, Matthew Hobbs, Jade L. Morris, Margaret A. Defeyter, Geir K. Resaland, Jim McKenna

**Affiliations:** 1Faculty of Health Studies, University of Bradford, Bradford, West Yorkshire BD7 1DP, UK; 2Centre for Applied Education Research, Wolfson Centre for Applied Health Research, Bradford Royal Infirmary, West Yorkshire BD9 6TP, UK; 3Center for Physically Active Learning, Faculty of Education, Arts and Sports, Western Norway University of Applied Sciences, 6856 Sogndal, Norway; geir.kare.resaland@hvl.no; 4School of Sport, Carnegie, Leeds Beckett University, Leeds LS6 3QT, UK; j.mckenna@leedsbeckett.ac.uk; 5School of Health Sciences, College of Education, Health and Human Development, University of Canterbury, Christchurch, Canterbury 8041, New Zealand; matt.hobbs@canterbury.ac.nz; 6Centre for Society and Mental Health, Department of Health Service and Population Research, Institute of Psychiatry, Psychology & Neuroscience, King’s College London, London WC2B 6LE, UK; jade.morris@kcl.ac.uk; 7Healthy Living Lab, Faculty of Health & Life Sciences, Northumbria University, Newcastle upon Tyne NE 7 7XA, UK; greta.defeyter@northumbria.ac.uk

**Keywords:** academic lessons, moderate-to-vigorous physical activity, whole-school, physical activity, physically active learning

## Abstract

Background: A large majority of primary school pupils fail to achieve 30-min of daily, in-school moderate-to-vigorous physical activity (MVPA). The aim of this study was to investigate MVPA accumulation and subject frequency during academic lesson segments and the broader segmented school day. Methods: 122 children (42.6% boys; 9.9 ± 0.3 years) from six primary schools in North East England, wore uniaxial accelerometers for eight consecutive days. Subject frequency was assessed by teacher diaries. Multilevel models (children nested within schools) examined significant predictors of MVPA across each school-day segment (lesson one, break, lesson two, lunch, lesson three). Results: Pupils averaged 18.33 ± 8.34 min of in-school MVPA, and 90.2% failed to achieve the in-school 30-min MVPA threshold. Across all school-day segments, MVPA accumulation was typically influenced at the individual level. Lessons one and two—dominated by maths and English—were less active than lesson three. Break and lunch were the most active segments. Conclusion: This study breaks new ground, revealing that MVPA accumulation and subject frequency varies greatly during different academic lessons. Morning lessons were dominated by the inactive delivery of maths and English, whereas afternoon lessons involved a greater array of subject delivery that resulted in marginally higher levels of MVPA.

## 1. Introduction

Globally, half of all children do not achieve the recommended threshold of 60 min of daily moderate-to-vigorous physical activity (MVPA), this rises to between 70% and 76% in European countries [1,2,3,4]. Moreover, levels of MVPA decline by approximately 4.2% per year from the age of seven [4,5,6], when formalised teaching commences in many westernised countries. To reverse this decline, effective physical activity interventions are required during childhood school years. Whole-school approaches to physical activity [7] are recommended in global [8,9] and country-specific policies [10]. However, meta-analyses reveal that the current approaches have little, if any, effect on total daily MVPA [11,12]. The recently developed Creating Active Schools Framework perhaps shines a light on why such failures occur, as many previous interventions have failed to address the multiple factors required to operationalise a whole-school physical activity approach [7]. It is essential that whole-school interventions take a behaviour change approach to address policy, mobilise stakeholders, create effective social and physical environments and implement physical activity across the seven available opportunities [7].

The majority of children attend school for six to seven hours per weekday, giving ample opportunity for intervention [13]. Highlighting the potential of schools, many national policies, including those of the United Kingdom (UK), recommend that schools provide all children with a minimum of 30-min in-school MVPA per day [10]. A recent systematic review [14] and meta-analysis [15] showed that in-school MVPA ranged between 16 and 61 min. However, 90% of children failed to achieve the 30-min MVPA threshold [14]. Such low levels of MVPA during school hours is likely to have a negative impact on pupils’ physical and mental wellbeing [16,17]. Of equal, if not greater, importance to the education system, inactive school environments could undermine academic performance [18]. Yet, while positive associations are being observed between academic performance and physical activity [19], interventions within-schools have failed to demonstrate benefits in any subject other than maths [20]. Such shortcomings might be attributed to the failure of programs to address the many factors required for the effective implementation and poor sustainability of current interventions [7].

Enhancing the intervention design requires a detailed understanding of MVPA accumulation during the school-day. Discretionary physical activity occurs= where children have control of their choices, such as break (recess) [21]. In compulsory segments, i.e., academic lessons, the activity is directed by others. The physical activity profiles of discretionary and compulsory segments have been highlighted in previous research [22,23,24,25,26], identifying break/lunchtime and Physical Education (PE) lessons as the most active in-school periods. Inactive and sedentary academic lessons often dominate the school day [13], meaning they offer an additional avenue for increasing in-school physical activity beyond the traditional focus on break [27] and Physical Education [28]. Five systematic reviews [29,30,31], two including a meta-analysis [32,33] support focusing on integrating more movement within curricular lessons, highlighting the potential of physically active learning (PAL) and classroom movement breaks (CMB) for increasing the levels of MVPA.

Increasing the effectiveness and implementation of PAL and CMB interventions requires a greater understanding of MVPA accumulation within academic lesson segments. While a growing evidence base exists for the school-level effects on MVPA during PE, break and lunch, limited understanding exists for academic lessons [23,26]. Compounding the issue, current segmented-day research combines all academic lessons into one homogenous segment. This approach likely masks that different lesson segments have varying physical activity profiles and underlying lesson constructs [25].

As lessons are based on different subjects, such as maths or PE, with differing physical activity profiles [34], they should be assessed as their own entity, not together. Such refinements in the current thinking would support intervention developers to target the most inactive lessons. In addition, understanding the delivery frequency of the different subjects with which movement could be integrated could promote further MVPA across the school day. The primary aim of the study is to investigate the impact of different lesson segments on MVPA accumulation in primary school children. A secondary objective is to identify the frequency and distribution of the different National Curriculum subjects across the different lesson segments.

## 2. Materials and Methods 

### 2.1. Participants 

The current study uses baseline data from the participants who took part in the Redcar and Cleveland physical active learning project [35]. Participants were recruited from Year Five (aged 9 to 10 years) classes across six primary schools in the North East of England. School Games Organisers approached the selected schools, forming a convenience sample. The proportion of children receiving free school meals across schools ranged from 31.8% for the most deprived to 2.4% for the least deprived (*M* = 16.4%). Institutional approval was received from Leeds Beckett Research Ethics Committee (reference: 37482). Following headteacher consent, parents and pupils were sent information letters to their home address via the school. All participants who provided consent and assent were included in the study.

### 2.2. Protocol 

Data collection took place in January, winter in England. Two schools were visited per day over three consecutive days. During the visit, pupils were provided with accelerometers and had their height (meters) and body mass (kg) measured. While accelerometers were being distributed, the lead researcher encouraged pupils to identify strategies to increase wear-time, e.g., “placing it next to my toothbrush”. Following the accelerometer briefing, the lead researcher explained the class-level activity diary to the teacher.

### 2.3. Physical Activity Assessment 

Physical activity levels were measured objectively, using a combination of GT1M and GT3X accelerometers (ActiGraph, Pensacola, FL, USA). While it is preferable to use the same accelerometer model, previous research has demonstrated the high level of compatibility of these two monitors in standardised and free-living activities [36,37]. Monitors were worn on the right hip, in alignment with previous school-based studies in children [25]. Participants were required to wear the monitors for eight consecutive days, (i) every day, (ii) throughout the day, except for sleeping and water-based activities, and (iii) to continue wearing the monitor, even if a day was missed. Monitors were set to record from 12 a.m. on day two, producing seven days of data collection while allowing for a minimum of a 12-h induction [38]. Data were collected in 15-s epochs. The epoch length was chosen alongside the selected cut-points. Utilising an epoch that differs from those validated in conjunction with the cut-points can lead to inaccurate outcomes [39,40]. Evenson [41] cut-points were used to determine time spent in sedentary time (0–25 counts 15^−1^), light physical activity (26–573 counts 15^−1^) and MVPA (≥574 counts 15^−1^).

Accelerometer data were downloaded using Actilife (version six, Pensacola, FL, USA) and then converted into AGD files prior to being analysed in Kinesoft (v3.3.75, Kinesoft, Loughborough, UK). Non-wear time was identified by a period of ≥60 min of zeros allowing for a period of 2 min non-wear time [5], with the total duration of these blocks representing non-wear time. Spurious values were identified as ≥30,000 counts. Valid-wear criteria were set at ≥3 school days [42,43] with a wear time ≥480 min per day. While 480 min is at the lower end of the wear criteria and may underestimate total daily MVPA levels [42], a significant proportion of school-based studies in children have utilised this threshold. Due to the focus on in-school physical activity, a longer wear-time was not deemed necessary for the current study. To confirm in-school wear-time for valid days, a further visual check of each accelerometer profile was undertaken. For segmented day analysis, independent segments were removed if < 100% wear time was identified through visual screening.

### 2.4. Teacher Diaries 

In-school segments were characterised using teacher diaries. Specific segments were extracted from the pupils’ accelerometer profiles using the windows function in Kinesoft. Teachers were requested to complete the diary immediately after each segment, noting timings for the start and end of the school day, lessons one, two and three, breaks and lunchtimes. In addition, teachers were requested to record different subjects that occurred within each lesson period [26]. The following segments were identified for all schools:In-school; the beginning of the first lesson to the end of the last lesson;Lesson one: start of school to the beginning of the first break (recess);Break: end of lesson one to the beginning of lesson two;Lesson two: end of the break until the beginning of lunchtime;Lunchtime: end of lesson two to the beginning of lesson three, includes time for eating;Lesson three: end of lunchtime until the end of school;Lesson time: total lesson time (lesson one + lesson two + lesson three).

### 2.5. Identifying Lesson Type Frequency 

Lesson frequency was the number of times a subject appeared within each of the different lesson segments (lesson one, lesson two, lesson three). These were totaled across schools. Teacher diaries established the day and the frequency with which the different subjects (maths, geography, English, history, science, languages, computing, music, art and design, PE, Personal, Social, Health and Economic education (PSHE) and design and technology) occurred in each lesson segment. These were coded in alignment with the Key Stage Two National Curriculum for England [44]. After a visual check of the teacher diaries, two additional lesson types were identified, as they occurred in two or more schools: assembly—a gathering of part or whole school for a special programme or communication of information; golden time—free time for pupils to self-select activity which was often used as a reward for hard work.

### 2.6. Anthropometry

Height: Secca 213 floor standing height measure (Seca Deutschland, Hamburg, Germany) and body mass: Secca 877 (Seca Deutschland, Hamburg, Germany) were measured in alignment with the “Assessing the Levels of Physical Activity and Fitness “ALPHA fitness testing battery protocol [45]. Prior to measurement, participants were asked to move jumpers and shoes, completing the measurements in trousers or a dress and a shirt. Body Mass Index (BMI) was calculated using weight (kg)/height^2^ (m). The British growth reference chart values were used to convert BMI into standard deviation scores (BMI z-score) while accounting for normal growth by age and gender [46]. Using BMI z-scores, children were classified as normal weight (<85th centile, BMI z-score < 1.04), overweight (85th to 95th centile, BMI z-score 1.04–1.639) and obese (95th centile, BMI z-score > 1.64). Biological maturity (maturity offset) was established using children’s age from peak height velocity (APHV) [47]. Due to not capturing sitting height, the standing height simplified equation was used for both boys and girls [47].

### 2.7. Data Analysis

Individual and school-level descriptive characteristics are presented as *Mean ± Standard Deviation (SD)* for all measured variables. Normal distribution was confirmed for all variables using the Kolmogorov–Smirnov test (*p* > 0.05). One-way ANOVA (Tukey post hoc) assessed differences between schools in baseline characteristics (i.e., age) and levels of MVPA across the whole day, in-school and all school segments (*p* < 0.05, *p* < 0.01, *p* < 0.001, *p* < 0.0005).

Separate multi-level models (unstandardised coefficients (*b*)) identified the intraclass correlations (ICC) and significant predictors of the MVPA across each independent segment (lesson, break, lunch, lesson one, lesson two, lesson three). Intraclass correlations identify “the proportion of the total variability that is attributable to the level two unit” [48]. Multi-level models account for the clustering of pupils within schools. Independent models were constructed with random intercepts for MVPA, across the different segments of the school day. Progression to random slopes resulted in output errors. Models were built using the recommended three-stage process [49]. First, only the level-two clusters (the schools) and outcome. Second, the level-one predictors were added and third, school-level predictors were added. All predictors were identified a priori through the previous literature and have been shown to affect the amount of time spent within each activity threshold during key segments of the school day [50]. School-level predictors included the percentage of pupils receiving free school means (FSM) and segment length. Pupil-level predictors included gender, maturity offset and BMI z-score. The significance of the changes in the models were assessed using -2 log-likelihood with significance accepted (*p* < 0.05). Residuals were normally distributed, and all analyses were conducted in IBM SPSS 21 (IBM, Armonk, NY, USA).

## 3. Results

### 3.1. Sample Characteristics 

Of the 149 participants (*M_age_* 9.91 ± 0.30 years), 122 (82%) returned valid accelerometer profiles (Table 1). No significant differences in measures were observed between 27 excluded participants and the final sample of 122 participants. For the final sample (*n* = 122), only height was deemed significantly different between the six schools (*F*_(5,121)_ = 3.90, *p* = 0.003; Table 1) and there was a higher proportion of girls within the sample compared to boys.

### 3.2. Average Weekday and In-School MVPA Accumulation 

On average, pupils accumulated 44.90 ± 17.04 min of MVPA per day. This resulted in 17.2% of pupils achieving 60 min of MVPA per day. While this varied greatly between schools (range 38.33 to 49.65) the differences were not significant (MVPA (*F*_(1,121)_ = 1.81, *p* = 0.116). The average duration of time spent in school was 378 min, ranging from 368 to 390 min. Pupils accumulated 18.33 ± 8.34 min of MVPA during school hours (school range 12.73 ± 4.89 to 22.18 min). Overall, 9.8% of pupils accumulated >30 min of in-school MVPA per day, ranging from 0% to 23.8% across schools (Figure 1). Across schools, sixty-five percent of pupils accumulated <20 min of in-school MVPA each day (range 43% to 94%), and 12.3% accumulated less than 10 min.

### 3.3. Segmented In-School MVPA Profiles 

Average total lesson duration was 312.5 min (range 305–325 min), of which, pupils spent 7.81 (±4.05) min in MVPA (Table 2). One-way ANOVA and post hoc analyses revealed great variability between schools (3.71 min to 10.94 min (*F*_(1,121)_ = 14.72, *p* <0.0005). Similarly, during break time (range 15–20 min), pupils accumulated 2.53 (±0.24) min of MVPA (15% in MVPA). One-way ANOVA and post hoc analyses revealed significant variability between schools (*F*_(1,121)_= 7.38, *p* < 0.0005). During lunchtimes (range 45–60 min), mean MVPA was 8.00 (±4.57) min (16% in MVPA). Again, one way ANOVA and post hoc analyses revealed large variability between schools (*F*_(1,121)_= 4.06, *p* = 0.002).

A greater proportion of variance was explained at the school-level for lesson time (*ICC* = 32.12) compared to break (*ICC* = 0.00), and lunch (*ICC* = 7.92) (Table 2). No individual- or school-level correlates predicted MVPA accumulation during lesson time. At break and lunchtime, gender and maturity offset predicted MVPA accumulation with gender-level effects at least two times greater than any other effect. Segment length was the only school-level correlate to predict an increase in MVPA at break.

### 3.4. Individual Lesson MVPA Profiles 

All lessons were relatively inactive, with only 2%, 2% and 4% of the available time spent in MVPA in lessons one, two and three, respectively. Pupils accumulated significantly more minutes of MVPA in lesson three (4.65 ± 2.86, *p* < 0.0005; range 1.90 ± 0.95 to 7.62 ± 2.54 min) compared to lessons one (1.75 ± 1.56; range 0.58 ± 4.33 to 2.71 ± 2.08 min) and two (1.41 ± 0.93; range 1.01 ± 0.59 to 2.64 ± 0.82 min). A greater proportion of variance existed at the school level during lesson three (*ICC* = 49.81) compared to lesson one (*ICC* = 9.18) and two (*ICC* = 9.38; Table 3). Segment length predicted time spent in MVPA during lessons one and two, but not lesson three. For every additional minute spent in lessons one and two, 0.04 min and 0.03 min of MVPA were accumulated, respectively, this effect was small. No individual level correlates predicted time spent in MVPA within any lesson.

### 3.5. Individual Lessons and Subject Frequency 

The frequency of subjects across all schools was totalled by lesson period (Figure 2). Maths and English were the most frequently delivered subjects, totalling 36 and 44 occurrences respectively. This equates to seven sessions of maths and nine sessions of English, per school, per week, on average. Maths (43%) and English (44%) were the most frequently delivered sessions in lesson one (87% of all observations). Assembly was the only other subject to appear in lesson one (13% of occurrences). In lesson two, once again, maths (38%) and English (52%) appeared with the greatest frequency, totalling 90% of all observations. History (3.3%), science (3.3%) and PSHE (3.3%) each occurred once. During lesson three, all subjects were included (maths, English, science, computing, art and design, design and technology, geography, history, languages, music, PE, PSHE, topic, assembly and golden time); with English and PE occurring most frequently.

## 4. Discussion

This study advances current knowledge by presenting a detailed understanding of MVPA accumulation and subject frequency during academic lessons and across the school day. Extending the evidence base, school-level variability in MVPA accumulation was three times higher for academic lessons, than at break and lunchtime. Uniquely, we demonstrated the heterogeneous nature of academic lesson segments, with a greater school-level effect for lesson three (afternoon), compared to lessons one and two (morning). In lessons one and two, both dominated by maths and English, levels of MVPA were low. In lesson three (pm), while MVPA levels were higher and a greater variation in subjects were delivered, still only 4% of time was spent in MVPA. Consistent with the previous literature [23], gender, BMI z-scores and maturity offset did not predict lesson time MVPA.

In-school MVPA was 18 min, aligning with the lower end of the range (16 to 61 min) presented in a recent systematic review [14], and below that (24.8 min) observed within a more recent meta-analysis using Evenson cut points [15]. Even though average levels of MVPA were lower than in previous studies, consistent with the previous literature, 90% of children in this study failed to accumulate 30-min of in-school physical activity per day [14]. Providing a more detailed understanding of in-school physical activity accumulation, 65% of children in this study accumulated less than 20 min of MVPA, with 12% accumulating less than 10 min. These novel insights suggest that only substantial whole-school improvements in physical activity will deliver the policy-based recommendations of 30-min of in-school MVPA per day.

In agreement with previous studies [23,25,26], pupils accumulated their lowest amount of MVPA during compulsory academic lesson segments, whereas discretionary periods, such as break and lunch, returned the most MVPA. In agreement with Fairclough et al. [23], ICCs from multilevel models revealed a ~24% greater school-level variance in MVPA accumulation during academic lessons, compared to break and lunch. This finding suggests that classroom teachers and, potentially, other classroom or school-level factors are highly influential in determining the amount of MVPA accumulated during lesson time. When separating out the different lessons, the school-level effect reduced considerably for lessons one and two, yet remained high for lesson three. It is plausible that such low school-level variance in lessons one and two was due to the dominance of sedentary teaching and learning approaches for maths and English. Yet, in the afternoons, lessons included a wide variety of subjects, some of which potentially aligned to more active teaching and learning methods. Just as all schools delivered maths and English in morning lessons, PE was always an afternoon lesson. This likely explains a large proportion of the MVPA accumulated during this time segment. Yet, it is important to note that MVPA levels were still low in afternoon lessons, especially as the segment included PE lessons. Therefore, it should be emphasised that all lesson segments are ripe for the increased integration of MVPA and more active approaches to teaching.

Previous systematic reviews and meta-analyses justify the need to integrate more physical activity within the delivery of academic subjects through PAL or CMB [29,30,31,32,33]. Specific insights generated in the present study, especially regarding the different academic lesson segments and subjects therein, will enhance future intervention design and implementation. The dedicated focus on maths and English in morning lessons, where the least MVPA was accumulated, suggests that intervention developers should primarily focus on these segments and subjects. This finding presents a paradox, as maths performance is shown to improve with physical activity, yet this was the least active part of the school day [20,51].

The present study adds more evidence to justify initial teacher training (ITT) providers integrating PAL within their courses [52,53]. The results from this study suggest building teachers’ initial capability and motivation to integrate more movement in high-frequency core-subjects may have greater potential to increase MVPA than a broad focus on all subjects. Having said this, ITT or qualified teacher training programs may encourage teachers to reflect on the teaching strategies they deploy within the broader curriculum areas as these appear to promote more physical activity. Further, to enhance teachers’ capability and motivation, schools also need to create opportunities for PAL. As seen in previous studies, schools should encourage using school gymnasiums and outdoor learning, while also providing physical resources to support the delivery of these lessons [7,54].

At a policy level, the study raises the need for a more holistic approach to education, where the physical and mental wellbeing of pupils is equal to their academic achievement. While the results are cross-sectional and limited to six schools in the North East of England, they provide insights into the current education landscape and the dominance of using sedentary approaches to teach core curriculum subjects. To move forward, further studies are required to confirm the results. Yet, when our results are shared at education conferences, the dominance of maths and English across morning lesson segments has not surprised audiences, providing further confirmation of their accuracy and relevance. As suggested within the Creating Active Schools Framework [7], this points to a need for national agencies responsible for health and education to align policies to promote the integration of health and well-being in the curriculum. Scandinavian countries, who lead the international field in this regard, have school environments that promote greater levels of physical activity than in those countries where this is not the case [55,56]. For example, in Denmark, schools are expected to provide 45-min of MVPA [57]. In Finland, the national “Schools on the Move” programme is funded by the Ministry of Education and Culture and organised by the Board of Education and regional state administrative agencies [58]. Both examples clearly demonstrate the national priority of physical activity in schools and the move to integrate movement within classroom lessons. Other countries are following suit, for example, the UK recommends “active lessons” within the latest PE and School Sport Premium Guidance [59]. To support schools, Daly-Smith et al. [52] provide a summary of the future directions for PAL implementation for policy, practice and research. The authors highlight the need for all three stakeholders to align to ensure all factors within the school-system facilitate PAL adoption and implementation.

### Strengths and Limitations

Multi-level models represent an important methodological innovation for addressing variability in outcomes with multi-site interventions, accounting for the clustering of pupils within schools. Yet, due to the smaller numbers of individuals (level-one) within a few schools (level-two), the results should be interpreted with caution; fewer than 20 units at level two may introduce bias [60]. However, a strength of the current study lies in using this innovative analysis to reveal the pattern of MVPA and subject frequency in academic lessons. It should be noted that while the sample size may be small, it may not be feasible to conduct exploratory analyses in larger datasets. Further, the MVPA and ICC outcomes agree with the previous literature [23,26], supporting the generalizability of the findings. However, for confirmation, large-scale studies should replicate the method with increased numbers of level one and two units. When comparing the results to the other literature, researchers should account for the methodology, specific geographical and socio-economic context and that data were collected in winter, when MVPA levels are typically lower than in the spring and summer [6,61].

While physical activity was collected using accelerometers, uni-axial count-based data was used for the analysis. Future studies may choose to use triaxial accelerometers and adopt raw accelerations for their analysis [62]. While the teacher diaries have provided novel insights into the lesson pattern and frequency of curricular subjects, more objective methods of data collection may provide more accurate insights and may wish to be adopted in future studies. Further, capturing the delivery timings of the individual subjects would have enabled the data for Physical Education to be extracted from the afternoon lesson segment. Due to the usually high levels of MVPA in PE [63], it is likely that these lessons were at least partly responsible for the higher levels of MVPA in lesson three. While undertaking this analysis would not drastically alter the message that all lessons, especially morning lessons, are inactive and ripe for intervention, future large-scale studies should look to address this limitation.

## 5. Conclusions

In summary, to our knowledge, this study is the first to reveal the wide variation in MVPA accumulation and subject frequency across primary school academic lessons. Morning lessons were dominated by sedentary maths and English, whereas afternoon lessons featured a greater array of subject delivery that resulted in marginally higher levels of MVPA. These novel insights strongly suggest that all lessons segments are largely inactive, with the greatest opportunity to expand in-school MVPA occurring within morning lessons, dominated by maths and English. The findings further emphasise the urgent need to integrate physical activity within all lessons, especially maths and English. To facilitate this, future lesson-based interventions may wish to use the Creating Active Schools Framework to operationalise the multiple factors required for a whole-school approach to PAL; these include school policy, physical and social environments and all five stakeholder groups. Further, to support schools to implement more physical activity in academic lessons, national policy may need to rebalance pupils’ academic achievement with their physical and mental well-being.

## Figures and Tables

**Figure 1 ijerph-18-00990-f001:**
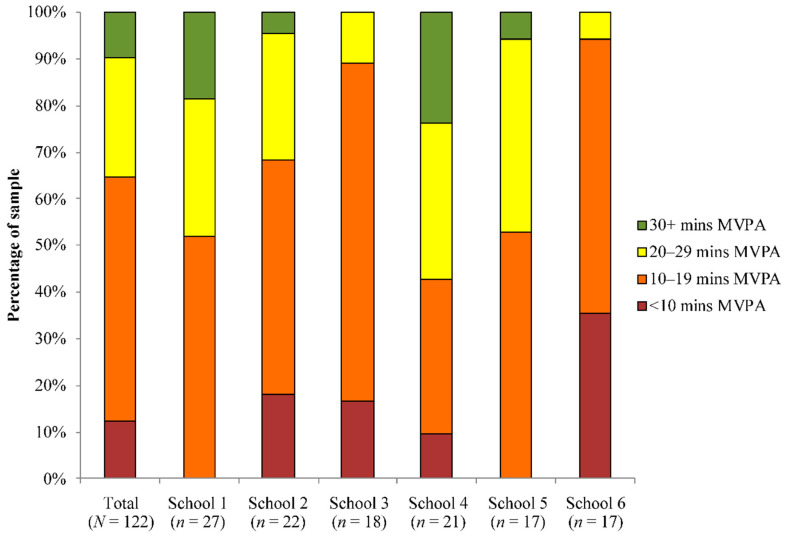
Differences between schools in the proportion of pupils who achieve 0–9 min, 10–19 min, 20–29 min and 30 + min of in-school moderate-to-vigorous physical activity per day; MVPA—moderate-to-vigorous physical activity.

**Figure 2 ijerph-18-00990-f002:**
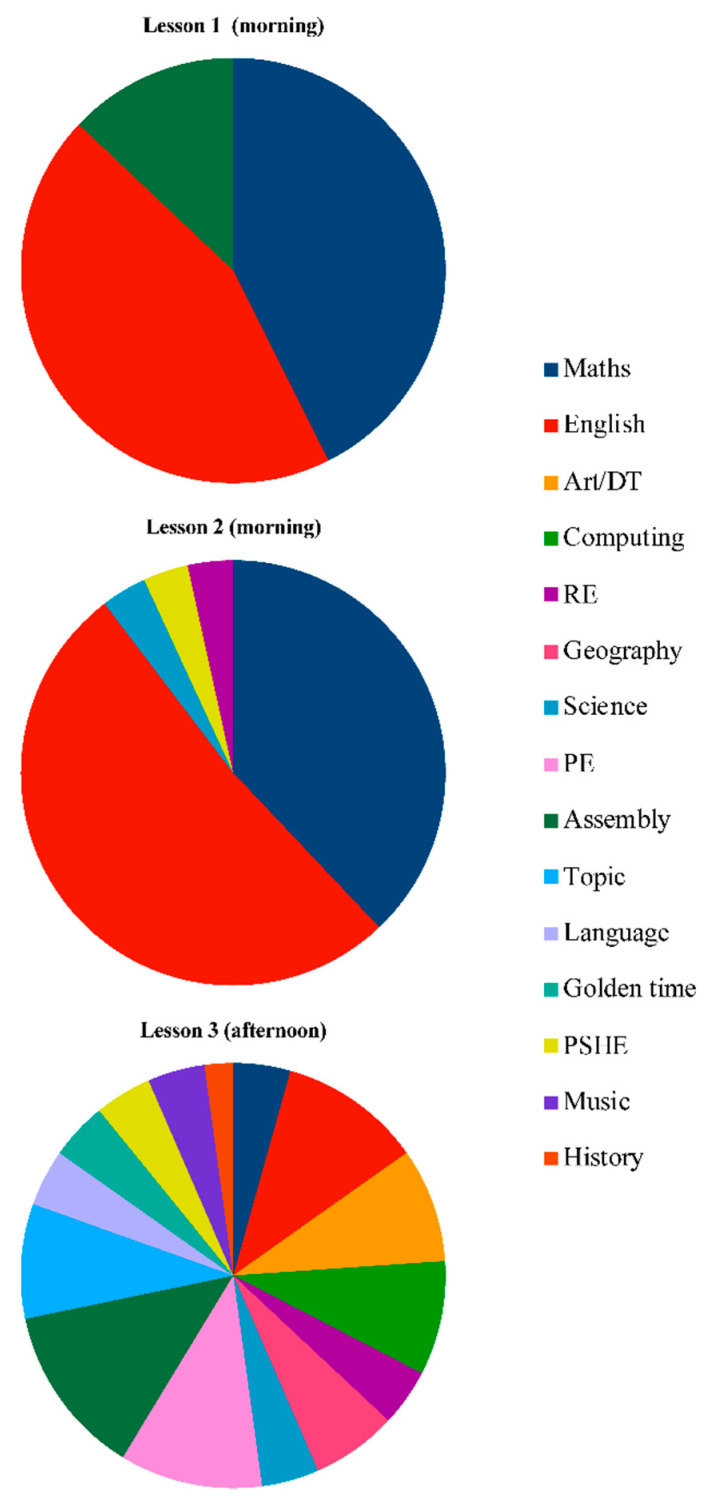
Total frequency of school subjects by lesson segment. (DT: design technology, PE: physical education, PSHE: personal, social, health and economic education, RE: religious education.).

**Table 1 ijerph-18-00990-t001:** Participant descriptive characteristics by school (*Mean ± SD* unless stated).

	Total (*n* = 122)	School 1 (*n* = 27)	School 2 (*n* = 22)	School 3 (*n* = 18)	School 4 (*n* = 21)	School 5 (*n* = 17)	School 6 (*n* = 17)
FSM (%)	15.8	2.4	15.4	24.1	31.8	14.36	10.2
Gender (%)							
Boys	42.6	44.4	45.5	38.9	52.4	35.3	35.3
Girls	57.4	55.6	54.5	61.1	47.6	64.7	64.7
Weight Status (%)							
Normal Weight	63.1	51.9	77.3	66.7	52.4	82.4	52.9
Overweight	17.2	18.5	13.6	16.7	23.8	5.9	23.5
Obese	19.7	29.6	9.1	16.7	23.8	11.8	23.5
Age (year)	9.9 ± 0.3	10.0 ± 0.3	9.9 ± 0.3	9.9 ± 0.3	10.0 ± 0.3	9.9 ± 0.3	10.0 ± 0.3
Height (cm)	140.6 ± 6.6 ^£^	143.8 ± 5.7 ^(5)^	139.1 ± 6.3	140.7 ± 7.4	140.6 ± 6.7	135.9 ± 4.5	142.4 ± 6.3
Body mass (kg)	37.4 ± 8.7	40.2 ± 9.1	35.5 ± 7.9	37.0 ± 5.8	38.8 ± 10.2	32.7 ± 6.9	38.8 ± 9.7
BMI (kg/m^2^)	18.8 ± 3.4	19.4 ± 3.9	18.2 ± 2.9	18.6 ± 2.3	19.5 ± 4.1	17.6 ± 2.6	19.1 ± 4.1
BMI z-score	0.7 ± 1.1	0.8 ± 1.3	0.6 ± 0.9	0.8 ± 0.8	0.9 ± 1.3	0.3 ± 1.0	0.7 ± 1.2
Maturity offset (y)	−2.3 ± 0.6	−2.2 ± 0.6	−2.4 ± 0.6	−2.3 ± 0.6	−2.4 ± 0.6	−2.4 ± 0.6	−2.1 ± 0.8
Accel wear time (mins)	717.4 ± 76.4	729.6 ± 73.9	720.2 ± 70.7	729.9 ± 89.4	690.6 ± 78.6	729.7 ± 86.8	702.7 ± 57.1

FSM: free school meals; BMI: body mass index; BMI z-score: standard deviation scores while accounting for normal growth by age and gender; weight classifications: normal weight < 85th centile, overweight 85th to 95th centile, obese > 95th centile; Significant difference between schools; £ = *p* < 0.001. Superscript number in brackets ^(5)^ identifies a significant difference from another school (*p <* 0.05). For example, in the table above ^(5)^ identifies a significant difference between School 1 and School 5.

**Table 2 ijerph-18-00990-t002:** Multi-level associations between pupil- and school-level predictors and moderate-to-vigorous physical activity across school segments.

	Lesson Time			Morning Break			Lunchtime Break		
	*b* (SE)	*p*	95%CI	*b* (SE)	*p*	95%CI	*b* (SE)	*p*	95%CI
Constant	−12.83 (16.30)	0.433	−45.12 to 19.46	2.93 (1.86)	0.123	−0.83 to 6.69	**16.14 (7.18)**	**0.04**	**0.88 to 31.39**
Pupil level variables									
Gender (ref boys)	−1.27 (1.12)	0.259	−3.50 to 0.95	**−2.85 (0.55)**	**<0.0005**	**−3.94 to −1.76**	**−6.09 (1.37)**	**<0.0005**	**−8.80 to −3.37**
Maturity offset	1.18 (0.88)	0.182	−0.56 to 2.92	**0.89 (0.43)**	**0.039**	**0.05 to 1.73**	**2.53 (1.06)**	**0.019**	**0.43 to 4.64**
BMI z-score	0.05 (0.27)	0.838	−0.48 to 0.59	−0.09 (0.13)	0.521	−0.35 to 0.18	−0.44 (0.33)	0.183	−1.10 to 0.21
School level variables									
FSM (%)	0.04 (0.10)	0.682	−0.20 to 0.29	−0.02 (0.02)	0.329	−0.06 to 0.02	0.01 (0.07)	0.878	−0.16 to 0.19
Segment length	0.08 (0.05)	0.123	−0.02 to 0.18	**0.39 (0.06)**	**<0.0005**	**0.24 to 0.54**	0.14 (0.10)	0.195	−0.09 to 0.38
Pupil level variance	10.04 (1.32)			2.45 (0.32)			15.24 (2.00)		
School level variance	4.75 (3.01)			0.00 (0.08)			1.32 (1.24)		
ICC	32.12			0.00			7.92		

FSM: free school meals; ICC: intraclass correlation; BMI z-score: body mass index standard deviation scores while accounting for normal growth by age and gender; SE: standard error; **Bold**: identifies significance (*p* < 0.05).

**Table 3 ijerph-18-00990-t003:** Multi-level associations between pupil- and school-level predictors and moderate-to-vigorous physical activity across academic lessons.

	Lesson 1 (Start to Break)			Lesson 2 (Break to Lunch)			Lesson 3 (Lunch to Finish)		
	*b* (SE)	*p*	95%CI	*b* (SE)	*p*	95%CI	*b* (SE)	*p*	95%CI
Constant	−2.57 (2.27)	0.271	−7.28 to 2.14	−0.36 (1.16)	0.756	−2.72 to 1.99	2.98 (13.88)	0.831	−24.97 to 30.94
Pupil level variables									
Gender (ref boys)	−0.01 (0.48)	0.977	−0.97 to 0.93	0.32 (0.27)	0.240	−0.85 to 0.21	−1.05 (0.71)	0.142	−2.46 to 0.36
Maturity offset	0.11 (0.37)	0.776	−0.63 to 0.84	0.19 (0.21)	0.370	−0.23 to 0.60	1.05 (0.55)	0.061	−0.05 to 2.14
BMI z-score	0.15 (0.12)	0.193	0.08 to 0.38	0.05 (0.06)	0.484	−0.08 to 0.17	−0.13 (0.17)	0.448	−0.47 to 0.21
School level variables									
FSM (%)	0.01 (0.02)	0.618	−0.05 to 0.07	0.00 (0.01)	0.807	−0.03 to 0.04	0.04 (0.09)	0.662	−0.19 to 0.27
Segment length	**0.04 (0.01)**	**0.026**	**0.01 to 0.07**	**0.03 (0.01)**	**0.007**	**0.01 to 0.06**	0.04 (0.11)	0.720	−0.18 to 0.26
Pupil level variance	1.88 (0.25)			0.58 (0.08)			4.00 (0.53)		
School level variance	0.19 (0.17)			0.06 (0.05)			3.97 (2.70)		
ICC	9.18			9.38			49.81		

FSM: Free School meals; ICC: Intraclass correlation; BMI z-score: body mass index standard deviation scores while accounting for normal growth by age and gender; SE: standard error; **Bold**: identifies significance (*p* < 0.05).

## Data Availability

The data are not publicly available as data sharing was not included within the original study ethics submission or participant consent form.

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
