# Peer review of "Moderate-to-Vigorous Physical Activity in Primary School Children: Inactive Lessons Are Dominated by Maths and English"

_ijerph, 2021, doi:10.3390/ijerph18030990_

Round 1
Reviewer 1 Report
The aim of this study was to investigate MVPA accumulation and subject frequency during academic lesson segments and the broader segmented. Accelerometers were used to assess physical activity of 122 students. Multilevel models examined significant predictors of MVPA during school day (across lesson one, break, lesson two, lunch, lesson).
The results of the research indicated that lesson one and two - (Math and English) - were less active than lesson three. Break and lunch were the most active segments.
The methodological side of the research is the strongest ponit of the manuscript. Apart from using accelerometers, the authors collected different information from teachers about the duration of the lessons, breaks and lunch time. They also performed antorpometric measurements.
The results of the research are obvious: Lesson one and two - dominated by Math and English - were less active than lesson three. Break and lunch were the most active segments. The weakest side of the manucript are conclusion. List of minor recommendations: -improve keywords list -shorten the part with the results -improve the conclusions - Figure 2 present in table form
In my opinion, the presented manuscript should be corrected according to the comments below:
We would like to thank the reviewer for their positive and constructive feedback. We have responded to all of the feedback and made the amendments where required.
Keywords:
- Too many keywords; in line 35 there are two statements mean the same moderate-to-vigorous physical activity; MVPA; one statement would suffice. The same note to physically active learning; PAL
Response: we would like to thank the reviewer for this comment. We have removed the abbreviations for PAL and MVPA.
Results:
- the test results are described in too much detail
Response: We have worked through the results section and removed any written detail that mirrors what is presented within the tables. In addition, any superfluous information has been removed. The remaining text and detail is required to enable the reader to have a complete understanding of not only the mean across schools, but also the high degree of variability that exists between and within schools. The author team feel this is an essential inclusion to provide the reader with a complete picture of the results.
-
improve the conclusions
Response: We have reviewed the conclusion and strengthened the takeaway messages for the reader, with a specific reference to practice and policy. The rewording of this section can be seen in lines 365 to 376.
- Figure 2 should be in the form of a table
Thank you for this comment. The author team has considered this request. We would, however, like to maintain figure 2 as a figure. Presenting as a figure enables the reader to capture the understanding of the data within one quick glance, by presenting this data in a table, the visual impact would be lost. The required detail behind the figure is included within the accompanying text. While this may extend the written results, the author team feel this is an acceptable balance. Further there are three tables and two figures within the paper, the author team would suggest this is an appropriate balance.
What are the main limitations of the study?
Response: A subsection heading has now been included within the discussion to draw attention to the strength and limitations section (line 340). This section currently highlights the main limitations as the overall sample size and sample size in relation to the mutli-level model, the limited geographic location and time of data collection. Three additional limitations have been added in regards to the use of accelerometers, teacher diaries and the inability to remove the PE data from the analyses. (See lines 353 to 363)
Reviewer 2 Report
jerph-1059811- review
Dear Authors,
Thanks for submitting this important manuscript.
Below please find my comments. I hope it can assist in improving the paper.
We would like to thank the reviewer for their detailed feedback. The points made have helped to improve the strength of the paper. The clarity in what was required has helped in ensuring we specifically address what the reviewer has requested.
Title:
Moderate-to-vigorous physical activity in primary school children: inactive lessons are dominated Maths and English – title not totally clear- unless I add the word “in* inactive lessons are dominant in math and English classes
Response: we have amend the title and included the word “by”
Abstract
First sentence- fail to reach 30 minutes per day? Per week? You probably mean per day but it is not clear
Response: we have added in the words “of daily” into the opening sentence. This now reads “A large majority of primary school pupils fail to achieve 30-minutes of daily in-school moderate-to-vigorous physical activity (MVPA)” (line 20)
Introduction
- Lines 39-40- would be better to base your information on the WHO recommendation for PA (file:///C:/Users/Owner/Downloads/9789241599979_eng.pdf) and data on levels of PA in European children on the HBSC study (http://www.hbsc.org/)
See for example a comparison international report that includes PA https://www.euro.who.int/en/health-topics/Life-stages/child-and-adolescent-health/health-behaviour-in-school-aged-children-hbsc/publications/2020/spotlight-on-adolescent-health-and-well-being.-findings-from-the-20172018-health-behaviour-in-school-aged-children-hbsc-survey-in-europe-and-canada.-international-report.-volume-2.-key-data
Response: Thank you for the suggestions we have included both of the suggested references into the opening sentence and have amended the detail accordingly to reflect the updated knowledge. (Lines 39 to 41)
- Lines 47-59- the discussion is focusing, if I understand correctly, on in-class (during academic lessons) physical activity time. It should be emphasized as recess PA time and physical education classes have positive impact of interventions. Another approach is introduction of additional PE classes and focusing of active recess. Where you discuss earlier the whole school approach, you can add a short explanation of what does it mean, and in this section refer to the various components and discuss the approach of interventions.
Response: Thank you for these comments. We have revised the introduction to include a short explanation of the whole-school approach, relating this to the failure of current whole-school approaches to address the many required components (Paragraph 1, lines 46 to 51). We have included further detail in regards to active breaks and PE in paragraph 3 (lines 69 to 74). We have not over emphasised the break and PE content as the focus of the introduction is to build towards justifying the need to have a more detailed understanding of lessons times. We do, however, agree with the reviewer as the need to make a brief mention of this.
- Line 60- typo error- school day, please edit
Response: Thank you, this has been amended.
- In lines 59-68 I see part of what I was referring to in my previous comment. I therefore suggest to re-structure these two sections to get the flow and to clarify,
- I do not fully understand the aim of the study (main one). Do you mean predictors? As in a cross- sectional study you can not look at affect on one factor on the other.
Response: thank you, this was useful feedback. On reflection, the aim of the study was confusing. We have now simplified this to “The primary aim of the study is to investigate the impact of different lesson segments on MVPA accumulation in primary school- children.” (line 87-89)
Methods and materials
- Lines 90-93 would be better in a separate section named ethical approval of similar
Response: We have checked similar articles in the journal (see Taylor 2017 reference) to ensure that we follow the journal style. It appears that the ethics detail is within the participants section. We have therefore not made this alteration. We are happy to amend this if required by the editor/ reviewer.
- How was sample size calculation done? Not clear why 5 schools were selected, school size (looking at clusters) etc
Response: Six schools were used within the current study. These schools were part of a physically active learning intervention study. A sentence has been included at the beginning of the 2.1 Participants section, detailing this. A supporting reference has been included that provides further detail on the sampling strategy for this study (see Morris, 2019). (Lines 93 to 94)
- I understand that accelerometers were used also at home. Why? If I look at study aim as presented at end on introduction- it looks only at in-school physical activity
Response: Total daily MVPA was captured and included in the results of the current study. This is to support the comparison of the study findings with other similar studies and overall physical activity levels.
- In the same line with previous comment- why was anthropometry taken if not part of study aims not even secondary?
Response: BMI z-score is used as a predictor variable in the mutli-level analyses.
- Line 166-167- all what is presented in this study is baseline- you have no intervention, the PA data is also baseline
- If your main analysis is the regression looking at predictors, it is important ton include the list of personal and school level predictors in the methods.
Response: thank you for noticing this, this was an oversight within the original submission. Predictors have now been included in the analysis section. (lines 187 to 189)
Results
- Table 1- not clear the marking of significance- is it between schools? Or? Please mark clearly and explain
Response: further clarification has been provided, with a specific example given, underneath table 1. (See lines 203 to 204)
- Figure 1- title not correct- you show distribution of PA time (not only 60 minutes) on each school
Response: Thank you for the suggestion, we have reviewed the title and taken a pragmatic approach due the difficulty in succinctly summarising the focus of the figure. The new suggested title is “Differences between schools in the proportion of pupils who achieve 0-9 minutes, 10-19 minutes, 20-29 minutes and 30+ minutes of in-school moderate-to-vigorous physical activity per day.” (Lines 215 to 216)
- Table 2- is the association between school level and individual level or between both to PA?
Response: Thank you, we have amended the title and altered the word ‘for’ to ‘and’ in both table 2 and table 3. This reflects the association between Pupil-level, School-level and MVPA. The titles also mirror ths labelling used in the Taylor (2017) article.
- Figure 2 lesson 3-as in lessons 1 and 2 no PE was included; PA time will be higher in this section. I would suggest removing all PE classes and calculate them separately to enable clear comparison between lessons 1+2 and 3. This is a major change in your results that might impact you discussion and recommendations, if found that once PE classes are removed, there is no difference between class1,2 and 3.
Response: We agree with the reviewer that this would have added a strength to the paper. However, we did not collect the specific timings of the lessons with the main lesson segments 1, 2 and 3. Therefore, we are unable to extract the specific PE data. Due to this, we propose the following; First, we have amended the method to clarify that the timings only relate to the overall segments, not the specific subject timings within these (line 136 to 140). Second, we have reworded the section in the discussion where we have previously acknowledged the fact that PE only appears in the afternoon segment and therefore this likely explains the higher MVPA scores in the lesson 3 (lines 294 to 300). Finally, we have added a further limitation to the strength and limitations sections to acknowledge that if this data had been collected, a more refined analysis of MVPA would have been possible (lines 357 to 363). Taking into account this limitation, the authors are confident with the key messages within the paper that classroom lessons are inactive and that the domination of Math and English during curricular lessons provides ripe opportunity for the integration of movement.
Discussion
- Are policies in Scandinavian countries promoting PA during academic class time? If yes, please include suggestions on how to incorporate it, as this is the core of your study and discussion.
Response: We have added in further information on the inclusion of PAL within wider PA implementation guidance in schools. In addition, we have drawn the readers’ attention to the Daly-Smith article that provides a recent summary of the required future directions in policy, practice and research for PAL. The text now reads “Both examples clearly demonstrate the national priority of physical activity in schools and the move to integrate movement within classroom lessons. Other countries are following suit, for example, the UK recommends “active lessons” within the latest PE and School Sport Premium Guidance [59]. To support schools Daly-Smith et al. [52] provide a summary of the future directions for PAL implementation for policy, practice and research. The authors highlight the need for all three stakeholders to align to ensure all factors within the school-system facilitate PAL adoption and implementation. (lines 333 to 338)
- I could find some study limitation hidden within your discussion; it will be better emphasized in its own section within discussion
Response: please see the previous response to reviewer one, point 5.
Conclusions
- Would add in line 337 “the first to our knowledge”, there might be another study that you didn’t come across
Response: we have added in the suggested wording. This sentence now reads “In summary, to our knowledge this study is the first to reveal the wide variation in MVPA accumulation and subject frequency across primary school academic lessons” (line 365)
Reviewer 3 Report
I received the manuscript entitled ‘Moderate-to-vigorous physical activity in primary school children: inactive lessons are dominated by Maths and English’ for expertise. The objective of this study was to investigate MVPA accumulation and subject frequency during academic lesson segments and the broader segmented school day. The results allowed the authors to highlight that MVPA accumulation and subject frequency varies greatly during different academic lessons. Overall, the paper is well written except for a few remarks that I will highlight below.
We would like to thank the reviewer for their positive feedback. We have addressed all of the points raised by the reviewer below and believe the inclusions accurately reflect what was requested and adds strength to the paper.
Remarks
Line 1: Remove the two extra dots
Response: Thank you, these have been removed.
Line 60: Replace “chool-day” with “school-day”
Response: Thank you, this has been amended
Lines 86-93: I would like to know how the authors determined their sample size?
Response: Please see the above response 13 to reviewer two who had the same question.
Line 165: ‘All data followed a normal distribution’. I would like to ask the authors to specify the test used to test the distribution of the data as well as the results (at least the p-value)
Response: This detail has now been included. The tests used was the Kolmogorov-Smirnov test and all p values were >0.05. This sentence now reads “Normal distribution was confirmed for all variables using the Kolmogorov-Smirnov test (p>0.05)” (lines 173 to 174)
Line 185: Only height … schools (…...; Table 1): In the table, we don't see 'height' but 'stature' I suggest to the authors to replace 'stature' with 'height' in table 1.
Response: Thank you for this suggestion, Stature has been replaced with height in table 1.(line 199)
Table 1: Superscript number in brackets is not (3) but (5)?
Response: Thank you for spotting this, we have amended the (3) to (5) in the table footnotes. (Line 202)
Round 2
Reviewer 2 Report
Then authors successfully addressed my coimments
Reviewer 3 Report
Good job